# The what, which, when, why and who of Off responses in the auditory system

**Jean-Marc Edeline**[1], **Robert C. Liu**[2]

[1]Paris-Saclay Institute of Neuroscience (Neuro-PSI, UMR 9197), CNRS – Université Paris-Saclay, Saclay, France

[2]Department of Biology, Emory University, O. W. Rollins Research Center, Atlanta, GA, USA

## Abstract

For decades, the Off responses noted in the auditory system from the cochlear nucleus up to the secondary auditory cortex have been largely ignored. Over the last few years, several studies have reinvigorated interest in Off responses in the auditory system and described their cellular mechanisms and the plasticity they display during various behaviours. While it is possible to design tasks where Off responses are necessary for an animal's behavioural performance, we posit that it is the interplay of On and Off responses that the brain uses more generally to enable sound feature perception, including sound duration and sound offset detection. Moreover, based upon close examination of the latencies and durations of published Off responses, we propose here that two different types of Off responses should be considered. We suggest that short latency–short duration responses participate in the neural coding of acoustic parameters whose estimates cannot be obtained on the sole basis of onset features. Meanwhile, long latency–long duration responses potentially reflect more cognitive variables associated with the learned use of a particular stimulus in a behavioural context. Proposing two varieties of Off responses draws attention to the need to better characterize post-stimulus firing in future studies to more definitively distinguish divergent mechanisms.

## Graphical Abstract

**Corresponding authors** Jean-Marc Edeline: Paris-Saclay Institute of Neuroscience (Neuro-PSI, UMR 9197), CNRS - Université Paris-Saclay, Centre CEA Saclay - Bât 151, 91400 Saclay, France. jean-marc.edeline@universite-paris-saclay.fr, Robert C. Liu: Department of Biology, Emory University, O. W. Rollins Research Centre, 1510 Clifton Rd NE, Atlanta, GA 30322 USA. robert.liu@emory.edu.
Author contributions
J.M.E. and R.C.L.: conceptualization; J.M.E. and R.C.L.: visualization; J.M.E.: writing-original draft; J.M.E. and R.C.L.: writing-review & editing; J.M.E. and R.C.L.: funding acquisition. Both authors have read and approved the final version of this manuscript and agree to be accountable for all aspects of the work in ensuring that questions related to the accuracy or integrity of any part of the work are appropriately investigated and resolved. All persons designated as authors qualify for authorship, and all those who qualify for authorship are listed.

Competing interests
The authors declare no competing financial interests.

In this article, we will first review 'What' different mechanisms are involved in the generation of Off responses at the sub-cortical and cortical level of the auditory system. Then, we evaluate 'Which' stimulus properties elicit Off responses at the different levels of the auditory system. Next, we specify the situations 'When' Off responses seem necessary to guide the animal's behaviour. We explained 'Why' the plasticity of these Off responses can be an important feature of a sound's neural encoding, and finally, based upon the latency and duration of the Off response, we consider two hypotheses for 'Who' within the larger neural network enables the generation of these Off responses. Created with BioRender.

### Keywords

auditory system; behavioural context; neural code; predictive coding; response duration; response latency

---

## Introduction

The long-standing interest in revealing neural codes – that is, for understanding the way neurons of the central nervous system represent external stimuli – has led to intense debates over the last 50 years (Azeredo da Silveira & Rieke, 2021; Eggermont, 1998, 2001; Gaucher et al., 2013; Huetz et al., 2011; Panzeri et al., 1999; Perkel & Bullock, 1968; Rieke et al., 1997; Wang, 2007). When recording techniques enabled sequences of action potentials emitted by individual neurons to be collected, researchers started using peri-stimulus time histograms (PSTH) to understand the relationships between stimulus parameters and the

occurrence of action potentials. Using such PSTHs, it was quite clear from studies of the visual system that spikes could occur at the onset, during and after the offset of visual stimuli. Although there has been considerable prior work on explaining how each of these response types have contributed to neural coding in the visual system, until recently the importance of firing after the end of the stimulus has been largely unappreciated in other modalities, including the auditory system.

In the visual system, the earliest descriptions of Off responses came from studies in the retina. Starting from Hartline (1938) to Barlow (1953), Kuffler (1953) and Hubel & Wiesel (1959, 1960), optic nerve fibres in the frog, cat or monkey were found to produce either pure onset ('On') or pure offset ('Off') responses when small circular areas of the retina were illuminated. Thus, the On and Off pathways are separated from nearly the beginning of the visual pathway. Functionally, these distinct pathways are crucial for motion detection in a large range of species from fly to rodent and to primate (e.g. see Behnia et al., 2014; Riehle & Franceschini, 1984; Schiller et al., 1986; Shechter & Hochstein, 1990). In early studies, as well as in subsequent ones (reviewed in Borst et al., 2020; Fu, 2023), the visual Off responses were considered as part of the 'normal' processing of visual information and were neither ignored nor assigned to a particular functional role. In fact, both experimental findings from different species and computer models have indicated that it is indeed the complementarity of the On and Off channels – and not one or the other – that allows direction selectivity and motion perception (Borst et al., 2020).

In the somatosensory system, peripheral dorsal root ganglion neurons have been classified as slowly adapting (SA) or rapidly adapting (RA) (Hunt & McIntyre, 1960a, 1960b), and the latter especially are known to exhibit Off responses when an indentation to the skin ends (Pei et al., 2009; Sur et al., 1984) – similar to the offset of visual stimuli in the retina. While early models of central somatosensory processing posited that SA and RA projections remain segregated up to and through the cortex, more recent studies indicate that these have already converged in primary somatosensory cortex neurons (Pei et al., 2009). In particular, cortical neurons with RA-like or mixed adaptation properties often exhibit Off responses (Emanuel et al., 2021; Pei et al., 2009; Simons, 1978; Sur, 1980). The function of this may be to participate in the population coding of the dynamical properties of mechanical stimuli, such as their duration, speed and/or directionality (Estebanez et al., 2012) – similar to what is observed in the visual system.

In the auditory modality, auditory nerve fibres only generate sustained tonic excitatory responses, but offset responses have long been described from the first central relay, the cochlear nucleus (Suga, 1964a, 1964b), up to the primary (Sołyga & Barkat, 2021; Sołyga & Barkat, 2019) and secondary auditory cortex (Chong et al., 2020). Two recent reviews have already summarized this expanding field of research and proposed functional roles for the Off responses observed in subcortical auditory structures (Kopp-Scheinpflug et al., 2018) and in auditory cortical fields (Anandakumar & Liu, 2022). The present review proposes first that it is only by integrating the properties of both On and Off responses that we can understand the functional role of Off responses in the auditory system. Second, when thinking about a potential functional role, we should consider that a difference probably exists between short duration Off responses and longer duration Off responses, as revealed

from disparate experimental designs that have progressively evolved from anaesthetized to passively listening animals to animals actively using the acoustic signals for performing a behavioural task. And third, it is also important to determine to what extent these responses are crucial for perceiving auditory stimuli, especially when we must extract meaningful auditory stimuli from the rich continuous stream of acoustic information reaching our cochlea.

Here, after reviewing the different mechanisms involved in the generation of the Off responses occurring at the brainstem and cortical level, the acoustic characteristics of the stimuli triggering these Off responses, and the circumstances where these responses seem crucial for behavioural performance, we will propose a broader coding importance of Off responses for sound feature learning and highlight that different types of Off responses potentially exist and have distinct functional roles. Both when resulting from network activity and from a combination of intrinsic properties and network activity, we should consider that they contribute significantly to different aspects of our daily perceptual performance.

### 'What' physiological mechanisms generate Off responses?

As previously reviewed across the entire auditory system (Anandakumar & Liu, 2022; Kopp-Scheinpflug et al., 2018), Off responses can be generated by a diversity of physiological mechanisms (Fig. 1), and, logically, one can wonder whether these different mechanisms occur at specific levels in the auditory system.

In the lower stages of the auditory system such as the dorsal cochlear nucleus, compelling evidence suggests that Off responses are generated by a post-inhibitory rebound mechanism (Rhode & Smith, 1986; Suga, 1964a, 1964b). That is, if a strong hyperpolarization occurs at sound presentation, at the end of the sound the membrane potential depolarizes activating T-type calcium channels (Fig. 1A1). This in turn generates large calcium currents and the emission of a burst of action potentials (review in Kopp-Scheinpflug et al., 2018). Note that a second scenario named 'post-inhibitory facilitation' (Fig. 1A2) slightly differs from the previous one by the presence of excitatory inputs which outlast the stimulus duration and facilitate the emergence of the Off responses.

At the cortical level, some results suggest that the mechanisms triggering Off responses differ from those operating at the brainstem level. Initially, it was suspected that the cortical On and Off responses are generated by different sets of synapses converging on the same cortical cell because it was noted that the frequency tuning of the On and the Off responses were often different, with sometimes one or two octaves of difference between the characteristic frequency (CF) of the On and Off responses (e.g. see Fishman & Steinschneider, 2009; Qin et al., 2007; Recanzone, 2000; Tian et al., 2013). In immature animals, though, there is often a perfect match between the CF of On and Off responses, and distinct frequency tuning between On and Off responses only emerges as a consequence of cortical development to promote cortical selectivity for upward or downward frequency sweeps (Sollini et al., 2018). Frequency modulated sounds are ethologically important, and retuning On and Off receptive fields to be more sensitive to them echoes the generation

of direction selectivity based on the On and Off patterns in the visual and somatosensory systems (Bensmaia et al., 2008; Mauss et al., 2017).

However, it is important to mention that differences between the CFs of the On and Off responses have also been reported in the auditory thalamus (He, 2002, 2003), indicating that at the thalamic level, two different sets of brainstem inputs may also promote the occurrence of On and Off responses. Results from intracellular recordings further suggest that thalamic Off responses can be caused by inhibition during the stimulus, and their significantly longer latencies indicate that the source of inputs might be different from that of the On responses (Zhang et al., 2008). Note that in some species lacking inhibitory interneurons in the auditory thalamus, the exclusive inhibitory input comes from the thalamic reticular nucleus, which may explain the longer latencies of Off responses in these cases (see also Cotillon-Williams & Edeline, 2003).

Today, the strongest indirect evidence supporting the hypothesis that cortical On and Off responses are driven by distinct sets of thalamo-cortical synapses converging on a given cortical neuron comes from a study using a forward masking protocol with pure tones (Scholl et al., 2010). It was found that pure On responses can display complete forward suppression but, with the same short inter-stimulus interval, Off responses are still present without any suppression (Scholl et al., 2010). This result was further supported by a later study that looked in a cell-type specific manner and found that the dynamics of On and Off responses exhibited different dependences on stimulus history (Olsen & Hasenstaub, 2022). In these studies, it was not possible to determine whether the two sets of separate inputs indeed corresponded to either two different thalamic inputs or to a thalamic and a cortical input. Nevertheless, a subsequent study using calcium imaging clearly showed that Medial Geniculate Body (MGB) terminals reaching primary auditory cortex (A1) express either On or Off responses, suggesting that the two sets of inputs likely come from thalamic origins (Liu et al., 2019). Interestingly, in a mouse model of gap-detection deficit, it was shown that the observed deficits in thalamic sensitivity to brief gaps in noise arise from reduced neural population activity following noise offsets, but not onsets. This suggests that gap detection deficits, which can be detected at the behavioural level, can arise from impairment of sound-offset channels at the thalamic level (Anderson & Linden, 2016).

We should keep in mind that other mechanisms have also been proposed for cortical Off responses. For example, 'inherited cortical Off responses' as part of a neural circuit can be based on the known existence of separated anatomical locations for the On and Off channels described within the auditory thalamus more than two decades ago (He, 2001). This hypothesis has been recognized by Solyga and Barkat (2021), but these authors suggested that the generation of Off responses is probably amplified at the cortical level as they detected a much larger proportion of 'onset–offset' responses at the cortical level (83%) than at the thalamic level (33%). In fact, Off responses are also more numerous and stronger in Anterior Auditory Field (AAF) than in A1I, and show a dependence on sound duration in AAF but not in A1 (Solyga & Barkat, 2021; Sołyga & Barkat, 2019).

In contrast with this straightforward inheritance from thalamus, Bondanelli and colleagues proposed that Off responses could be single-cell signatures of recurrent interactions

emerging within the cortical network (Bondanelli et al., 2021, Fig. 1C). As detailed later, in relation to this hypothesis Anandakumar and Liu (2022) have suggested that depending on the behavioural context, the Off responses can display different temporal decays. Thus, over the last years, converging results from different laboratories have suggested that the mechanisms of the origin of Off responses can fundamentally differ between the auditory brainstem and the auditory cortex: almost pure post-inhibitory rebound is suspected to operate at the brainstem level whereas more complex synaptic and network activity is suspected to operate at the cortical level.

In fact, the situation is likely even more complicated because other results also suggest that these two different mechanisms co-exist at the cortical level. For example, during a pairing protocol between a pure tone and a manipulation of post-synaptic activity aimed at suppressing On responses, it was sometimes observed that Off responses either emerged or were strongly amplified by the suppression of evoked on activity. For example, the cortical cell in Fig. 2A (from Cruikshank & Weinberger, 1996) displayed robust responses when activated by pure tones, but when the evoked On responses were prevented by simultaneous ejections of juxta cellular negative currents, this eliminated the tone evoked On response and led to robust responses at current offset. Other unpublished results (Fig. 2B) show that the Off responses emerge only when the hyperpolarizing current is strong enough to prevent the On response and that it is not a function of the sound duration but a function of the duration of the hyperpolarizing current (Cruikshank, 1997). This suggests that the cortical neurons could be driven to emit Off spiking via intrinsic post-inhibitory mechanisms rather than requiring synaptic excitation at sound offset. Consistent with this possibility, GABA application before sound presentation can consistently prevent the occurrence of On responses and lead to the emergence of Off responses (Fig. 2C–F, unpublished data from Manunta & Edeline, 2004). In fact, these data confirmed initial observations based on intracellular recordings where cortical cells displaying Off responses showed hyperpolarization during tone presentations (Volkov & Galazjuk, 1991; Volkov & Galazyuk, 1992).

In summary, whereas at the brainstem level the mechanisms allowing the emergence of Off responses mainly rely on post-inhibitory rebound (or post-inhibitory facilitation), it seems that a diversity of mechanisms potentially co-exists at the cortical level.

### 'Which' stimulus properties elicit Off responses at the different levels of the auditory system?

In many cases, Off responses were elicited by pure tones when testing the frequency tuning of neurons in the brainstem or thalamo-cortical auditory system. At the thalamic level, both the sound frequency and its intensity were investigated to determine whether they impact the presence of Off responses (He, 2001, 2002). At the cortical level, the sound intensity seems to play an important role as Off responses were often prominent at loud intensities (i.e. above 50 dB Sound Pressure Level, SPL) even when On responses can be detected for the same neuron at lower intensities (10 dB SPL, see Fig. 1*F* in Sołyga & Barkat, 2019). In area AAF, the strength of the Off response was a function of the sound duration and of the inter-stimulus interval (ISI): the longer the sound and the ISI, the stronger was the

Off response, with a similar effect observed for harmonic tones (see Fig. 3 in Sołyga & Barkat, 2019). However, note that the reverse was reported in the area A1 of awake mice: the Off responses elicited by noise bursts were clearly decreased with the duration of the noise bursts (Fig. 1C,D in Li et al., 2021). Lastly, the falling ramp used at sound termination seems to be crucial to elicit Off responses in AAF: a fast/abrupt ramp (0.01 ms) triggered more robust Off responses than a slow (10 ms long) descending ramp (see Fig. 2a,b in Solyga & Barkat, 2021). Slight differences exist with the results reported from layer 5 primary auditory cortex (A1) by Li et al. (2021). Whole cell recordings from layer 5 A1 neurons never showed pure synaptic Off responses; they all showed On–Off synaptic responses with offset responses weaker than the onset responses in terms of both excitatory and inhibitory currents. The Off responses of inhibitory PV and SOM interneurons were also weaker than their On responses and generally the Off responses evoked by pure tones were weaker than by broadband noise.

The advantage of a wider sound spectrum is further suggested by a recent study where Off responses to click trains were described in A1 of non-human primates (Song et al., 2024). In this study, their occurrence was also a function of both the inter-click interval and of the number of clicks in the trains: Off responses tended to occur for short inter-click intervals (less than 32 ms) and for a minimum of 8–16 clicks in the trains, potentially as a result of inhibitory mechanisms developing at particular rates of stimulation (see Fig. 3F in Song et al., 2024).

Lastly, natural stimuli with complex and sometimes broader band frequency trajectories, such as conspecific vocalizations, can trigger Off responses, especially in higher-order areas. For example, Chong et al. (2020) reported selective Off responses in both core and secondary (A2) auditory cortex of female mice. That these were elicited only by a small subset of ultrasonic vocalizations (USVs) over the 36 tested calls suggests that these responses are sensitive to the spectrotemporal structure of sounds and can be useful to discriminate the identity of the different calls. Indeed, by fitting the peak frequency of the vocalizations to sinusoidal and linear frequency modulated tones, the authors observed neural tuning around acoustic parameters (especially the amplitude of the frequency modulation, Fig. 3). They also noted that Off responses can even be tuned around the spectrotemporal parameters of non-ethological sounds (see Fig. 3B in Chong et al., 2020), suggesting they could play a more general role in signifying the occurrence of specific acoustic patterns that unfold over time *after* sound onset. Such an ability may not be very important for pure tones since no new spectral information occurs after sound onset, but it could be essential for discriminating spectrotemporally complex natural stimuli. This is further supported by studies of higher-order auditory–vocal motor areas in the songbird, such as area HVc, where some phasic responses occurring at particular syllables can display characteristics of Off responses (see Fig. 2 of Del Negro et al. (2005) and Figs 2 and 3 of Huetz et al. (2006)).

### 'When' are Off responses needed to behaviourally respond to sounds?

Over the last decade, several paradigms were specifically designed to reveal to what extent Off response can be crucial for guiding an animal's behaviour. In one of the first studies

(Weible et al., 2014), authors tested the consequence of silencing auditory cortex during a gap-in-sound detection task. They reported that the performance in gap detection was decreased when the auditory cortex was inactivated during the task. However, what remains unclear is whether the performance decrease was the consequence of suppressing the pre-gap sound's Off response or the post-gap sound's On response, which the work did not differentiate (and are likely conflated).

In fact, the most parsimonious explanation for the role of Off responses in the central auditory system is that they are necessary for signalling the end of an already present acoustic stimulus. In an elegant study, the authors designed a task where mice had to lick at the end of a sound of different durations to obtain a water reward (Solyga & Barkat, 2021). Once the mice mastered the task, the authors optogenetically activated the PV$^+$ cells in the auditory area AAF (an area showing more Off responses than A1) and looked at the consequence of preventing Off responses in non-PV$^+$ cells. Comparing trials when the PV$^+$ cells were activated to those when they were not, revealed that behavioural performance was slightly but significantly reduced and the reaction time was significantly longer. These data suggest that, in AAF, neuronal activity following sound termination is used by the animal to detect sound termination. Indeed, a linear classifier indicated that Off responses can be informative of the animal's decision suggesting that AAF Off responses impact behavioural performance. Notably, though, the classifier performance was the same when trained on On responses, pointing to the importance of having On responses before Off responses to detect sound duration.

In another study, the constraints on detecting sound termination were pushed further: mice were trained to lick for water reward for a particular noise duration (e.g. 300 ms) and not to lick to another duration (e.g. 50 ms; Li et al., 2021). Mice could learn to discriminate several duration combinations even if they were less effective in discriminating similar durations (e.g. 50 *vs.* 80 ms). Optogenetic methods were used to block the On and Off components of the cortical responses during several portions of the evoked activity. Blocking the phasic Off response (but not the delayed Off responses) produced a significant reduction of correct licking trials. Again here, blocking the On response alone also reduced the percentage of correct licking trials by about half suggesting that the presence of both phasic On and phasic Off responses is necessary for the detection of sound duration which echoes the complementary role of On and Off responses in processing visual and somatosensory information (Borst et al., 2020; Estebanez et al., 2012). Furthermore, optogenetic stimulation performed in a short temporal window after sound termination, which enhanced the phasic Off responses of pyramidal neurons, facilitated learning, that is, the correct detection of sound duration. Such data strongly suggest that A1 Off responses contribute to the perceptual recognition of sound duration. Unfortunately, so far, we do not know whether similar effects could also have been promoted by optogenetically manipulating subcortical Off evoked responses, which could be a direction for future research.

### 'Why' are cortical Off responses an important feature of a sound's neural encoding?

The studies reviewed above show that auditory cortical Off responses encode acoustic information that animals can be trained to use in behavioural tasks that release rewards.

Rather than assigning a specific function only for marking sound termination, though, we argue that Off responses can be useful more broadly in auditory coding. In particular, Off responses – whether they are generated by intrinsic, circuit or network mechanisms – likely serve more generically as a neural substrate to improve behavioural responsiveness to any acoustic feature that gives rise to them. The fact is that not all neurons fire Off responses at the end of a sound, so there must be some tuning to acoustic parameters, as discussed in the above section. In this sense, Off responses are similar to On responses, and as parameters like the frequency of a sound are varied, Off firing may appear only for specific ranges of frequencies (Fishman & Steinschneider, 2009; Recanzone, 2000; Scholl et al., 2010). Consequently, just as learning can enhance On responses in paradigms where acoustic features at sound onset are informative about rewards, it might be the case that learning can also enhance Off responses if they provide predictive value.

If this hypothesis is correct, then we should expect plasticity in Off responses after learning experiences that depend on the sound features encoded by Off responses. This is in fact what has been reported in two studies thus far. Chong and colleagues provided the first demonstration of experience-dependent Off plasticity by showing that its prevalence in A2 in response to mouse pup USVs increased in pup-experienced dams compared to pup-inexperienced virgins (Chong et al., 2020). Intriguingly, these Off responses were so-called 'Off-only' because neuronal activity was often suppressed during the sound itself (e.g. similar to Fig. 1B and E in Anandakumar & Liu, 2022), which suggests that they might arise from plasticity that refines post-inhibitory mechanisms akin to that observed in the brainstem (Fig. 1A) and cortex (Fig. 2). In fact, using electrophysiological methods, they reported that, in A2, the On response evoked by the USV weakened in experienced mothers, while the tuning of Off responses to frequency modulations in USVs shifted. This shift allowed neural firing to better discriminate between pup and adult USVs whose frequency trajectories changed differently over time – an acoustic feature for which Off firing that integrates over the preceding sound trajectory is well positioned to encode.

A similar result was also observed in a calcium imaging study (Lee & Rothschild, 2021) in the mouse A1, albeit using stimuli that were at least an order of magnitude longer than the USVs (several seconds *vs.* less than 100 ms). They operantly trained mice to recognize a particular sequence of juxtaposed pure tones (no gap) of different frequencies. Instead of the increase in Off response prevalence found in A2 after learning the sound sequence's behavioural relevance, the authors reported that the magnitude of Off firing actually increased, and was sustained when measured after learning. Even though these were longer stimuli (by at least a factor of 10), in both this case and for the shorter natural USVs, Off firing is much better suited than On firing to provide the critical discriminating information about the specific, dynamic frequency trajectory that is behaviourally relevant.

These works show the malleability of Off firing in the auditory cortex, but the mechanism for this plasticity, and whether it is similar to that for On responses, remains unclear. One intriguing but unconventional possibility is that the intrinsic and circuit mechanisms responsible for Off firing at the cortical level, discussed above, are altered by inhibitory plasticity (Vogels et al., 2013), which is seen at the cortical level when stimuli become behaviourally relevant through experience. In particular, stronger pup USV-evoked

suppression of neural firing has been reported in core mouse auditory cortex after females gain pup-care experience (Galindo-Leon et al., 2009; Lin et al., 2013; Shepard et al., 2016), and we speculate that this may lead to more neurons generating a post-inhibitory rebound within the auditory cortex (Fig. 1). So, the mechanisms described in Fig. 2 might be acting during learning situations to increase the probability of occurrence and/or the strength of Off responses. As that signal propagates within the cortical network, it might then increase the prevalence of Off firing at the cortical level. Although correlational at this point, plasticity in both inhibition and Off prevalence has been seen in maternal mice for natural pup USVs, and even for synthetic sounds paired with pups (Robert Lui's unpublished data), but a test of their direct causal relationship has not yet been carried out.

### 'Who' originates cortical Off responses – top-down, bottom-up or both?

Beyond a change in the prevalence of cortical Off responses, the maternal experience model in mice also uncovered plasticity in the temporal dynamics of Off spiking (Anandakumar & Liu, 2022). Pup call-evoked Off responses recorded in mothers were significantly longer in duration than those of pup-naïve virgin females, though latencies were statistically unchanged (see Fig. 6 in Anandakumar & Liu, 2022). This observation raises the question of whether the duration of Off firing might reflect differences in function depending on whether the evoking sound features have a behavioural meaning that informs subsequent motor actions. Indeed, we suspect that early/short duration Off firing – which is similar to the classical phasic pattern described for decades for the On responses at various levels of the auditory system – primarily encodes sound features (e.g. frequency, frequency modulation) in a reliable bottom-up fashion (Fig. 4A). As such, they reflect a neural code operating to signal the occurrence of stimulus parameters that might not be fully encoded by On responses.

In contrast, late/long duration Off firing (Fig. 4B), which is often seen in active listening or behavioural tasks – may not be strongly linked to specific stimulus parameters but to the potential use of this stimulus for behaviour. Indeed, sustained firing during a hold period between stimuli in animals performing a delayed-match-to-sample task is also a form of Off firing, which appears to play a role in working memory in mice (Yu et al., 2021). Notably, the latency of such long duration Off responses may still be relatively short (Fig. 4C), as seen in the case of mouse dams responding to randomly interleaved pup and adult mouse calls (Anandakumar & Liu, 2022), which we believe indicates the response's acoustic sensitivity. Hence, after a sound's behavioural meaning has been learned, a longer duration Off response could reflect the ability of the acoustic features to generate an echoic memory, which differs from working memory in that it can present even when animals are just passively listening instead of engaged in an active task (Cooke et al., 2020).

Finally, there are recent suggestions that Off period firing can have both long latency and long duration in the auditory cortex during an early phase of acquiring the meaning of a task-related sound (Drieu et al., 2025). The authors reported that while animals were learning a Go/No-Go task, within a few tens of trials from the onset of training, before they became experts, excitatory neurons showed heightened calcium signals during the Off response window, but primarily during hit trials. They argue that this represents an

instructive reward prediction, highlighting more cognitive factors that play a role in auditory cortical activity. We emphasize that even though those authors do not refer to this as an Off response, they do occur in a time period when the Off firing reviewed here would normally happen. That said, we speculate that this could be evidence of distinct forms of Off firing that need to be differentiated based on latency and duration, which future studies should more carefully document in their characterization of Off responses. Obviously, special care is needed to understand whether these neural activations are related to more top-down cognitive processes linked with the behavioural meaning of the stimulus or if these long latency/long duration Off responses reflect (i) an efferent copy of a motor command reaching the auditory cortex, or (ii) the planning of a motor behavioural response to be executed a few hundred milliseconds later. Distinguishing between these different possibilities is of crucial importance if one wants to understand the different functional roles of Off responses.

## Conclusions

After they were ignored for decades, there is now convincing evidence that Off responses are crucial for the behavioural detection of sound offset and for discriminating between different sound durations. Several results have clearly suggested that in specifically designed behavioural tasks, both On and Off responses participate in the neural mechanisms underlying duration perception. Temporal processing is crucial for auditory perception in general because natural sounds are constantly modulated over time in terms of spectral content and temporal envelope. Therefore, we need as many temporal markers as possible to segregate acoustic objects from the natural acoustic streams. The fact that Off responses represent an additional way to generate neural markers of particular events while listening to natural acoustic streams is an advantage compared to the situation where the auditory system could only analyse the flow of acoustic information based on On responses.

By now, we hope it is clear that ignoring Off responses could miss an extremely important component of how complex sounds that become behaviourally relevant are encoded in the auditory system. In the history of sensory neurosciences, the use of PSTH- based methods highlighted neural coding based on action potentials occurring just after, or within a few hundreds of milliseconds after, stimulus onset. This is incomplete because the neurons of the central auditory system have no information about the time of stimulus onset. For decades, this methodological bias has led the auditory community to focus just on On responses. Instead, we expect that more unbiased views of studying the neural code at play during, *but also after*, auditory stimuli, will Offer deeper insights into how the acoustic and cognitive dimensions of natural sounds are processed in the central auditory system.

## Supplementary Material

Refer to Web version on PubMed Central for supplementary material.

## Acknowledgements

We wish to thank Dr Scott Cruikshank for sharing his published and unpublished data with us. We thank Dr Valérie Ego-Stengel and Dr Luc Estebanez for clarifications about Off responses in the somatosensory cortex.

**Funding**

J.M.E. was supported by grants from the French Agence Nationale de la Recherche (ANR) (ANR-14-CE30–0019-01). R.C.L. was supported by a grant from the United States National Institute of Health (NIH) (R01DC008343).

## Biographies

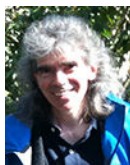

**Jean-Marc Edeline** is a team leader at the Paris-Saclay institute of Neuroscience (NeuroPSI), Saclay, France. His current research focuses on the neural mechanisms allowing perception of communication sounds in noise by combining behavioural experiments with electrophysiological recordings at several levels of the auditory system

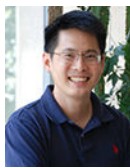

**Robert C. Liu** is a Full Professor in the Department of Biology at Emory University, Director of the Neuroscience Graduate Program and Associate Director of the Centre for Mind, Brain and Culture. His Computational Neuroethology Laboratory focuses on understanding the neurobiology and neural coding underlying social information processing and learning by using computational, electrophysiological, optogenetic, neurochemical and behavioural methods to investigate natural social behaviours like acoustic communication and pair bonding in rodents.

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

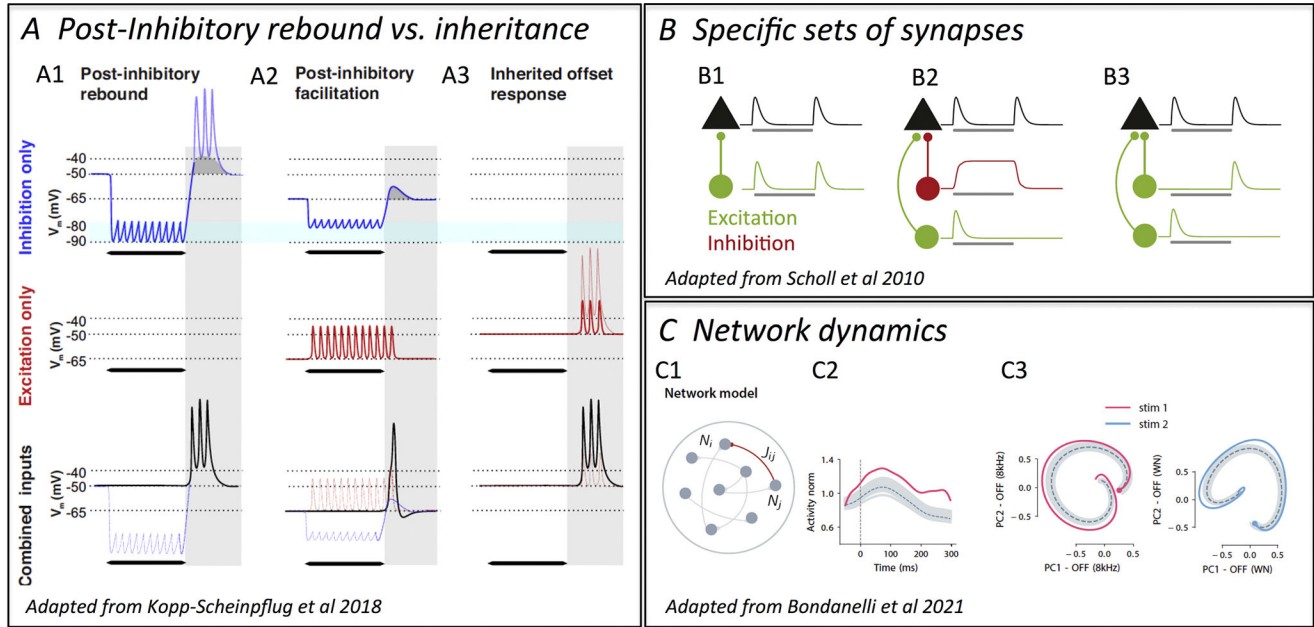

**Figure 1. Different mechanisms proposed for the occurrence of Off responses**

*A*, at the brainstem level, different hypotheses have been tested ranging from simple post-inhibitory rebound (*A1*), post-inhibition facilitation (*A2*), and inherited offset response (*A3*). During the sound (black bar), inhibitory (blue trace) or excitatory (red trace) inputs can combine (black trace) to generate spiking after sound termination through various cell-intrinsic properties or inheritance from separate synaptic inputs. Modified from Kopp-Scheinpflug et al. (2018). *B*, three types of scenarios for the synaptic mechanisms underlying On and Off responses in auditory cortex neurons. The spiking during On and Off periods of a cortical neuron can be produced either by excitatory (green) or by inhibitory (red) presynaptic neurons. The grey bar indicates the sound duration. On and Off responses can be generated either by the same set of excitatory synapses (*B1*); by a rebound from sustained synaptic inhibition (*B2*); or by different sets of presynaptic excitatory inputs (*B3*). Modified from Scholl et al. (2010). *C*, cortical recurrent network dynamics may also explain the emergence of Off responses. *C1*, a recurrent network model whose neural firing rates and connection strengths $J_{ij}$ between neurons $i$ and $j$ define the network state. *C2*, distance from baseline of the population activity vector during the OFF response to one example stimulus (red trace). Grey traces correspond to the normalized projection of the population activity vectors obtained from fitting the network model to the OFF response to a single stimulus and generated using the fitted connectivity matrix. *C3*, coloured traces: projection of the population OFF responses to two different stimuli on the first two principal components. The initial condition is indicated with a circle. Modified from Bondanelli et al. (2021).

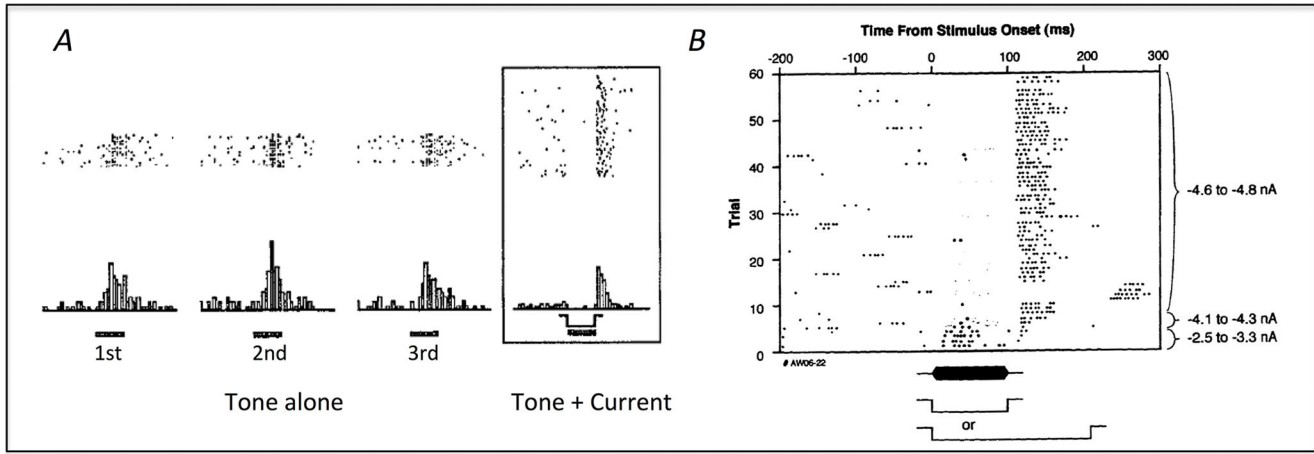

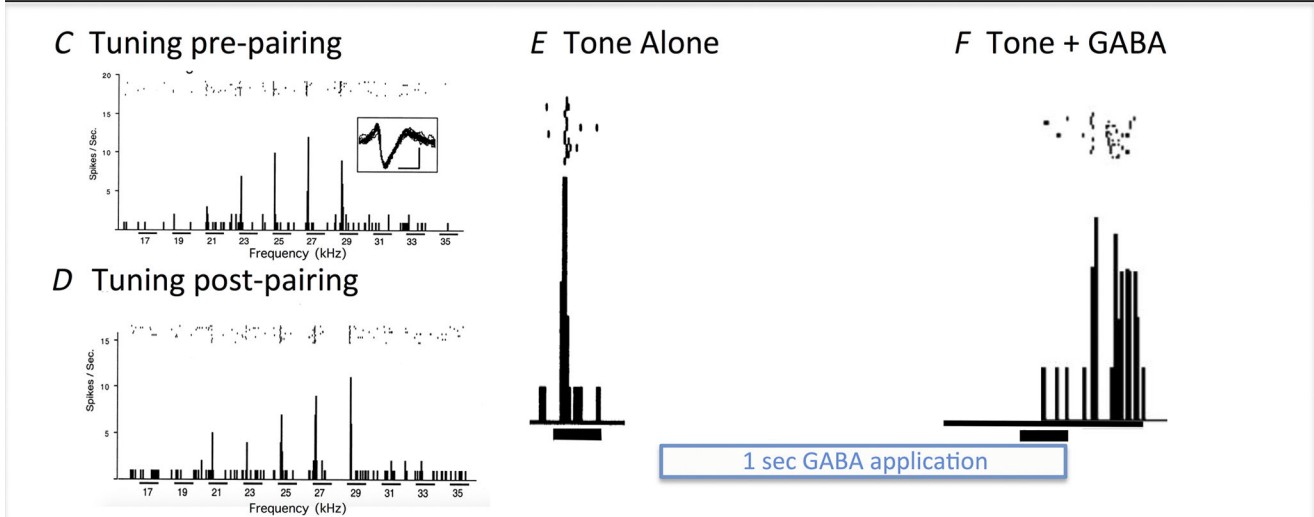

**Figure 2. Emergence of cortical Off response after strong suppression of activity**

*A*, during a Hebbian pairing protocol, a pure tone was first presented alone during three blocks of 20 trials (noted 1st, 2nd and 3rd) then a negative current was systematically applied at each sound presentation and prevented the neuron from emitting AP (Tone + Current). This almost complete suppression of tone-evoked activity was followed by a strong Off response (modified from Figure 7 in Cruikshank & Weinberger 1996, copyright 1996 Society for Neuroscience). *B*, the emergence of Off responses requires that the negative current totally blocks the On responses. In this example, the initial value of injected current (2.5–3.3 nA, bottom) was not sufficient to prevent the tone-evoked On response. When the current was increased to 4.1–4.3 nA, Off responses started to emerge and were even more pronounced when the current was set to 4.6–4.8 nA. Note that when the pulse duration was increased, the occurrence of the Off response was also delayed (first four trials at the 4.6–4.8 nA level). By permission of the author, from Cruikshank 1997, unpublished data). *C–F*, Hebbian pairing protocol between a pure tone and a pulse of GABA led to emergence of Off responses. This neuron exhibited robust On responses during the pre- and post-pairing tuning curve evaluation (*C* and *D*). On responses were also observed when a pure tone alone was presented (*E*), but they disappeared during a pairing protocol between the tone and a repeated micro-iontophoretic application of GABA in the vicinity of the cell. During this

pairing, Off responses were detected at the end of the sound presentation (unpublished data from Manunta & Edeline 2004).

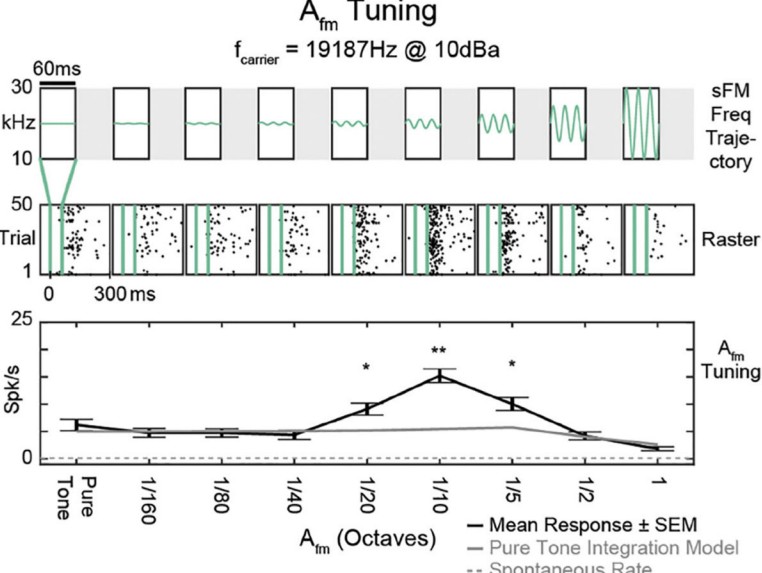

**Figure 3. Tuning of Off response to frequency modulation**

Tuning of an example single unit (SU) to sinusoidally frequency modulated tones centred around its best frequency (BF) of 19,187 Hz. Top, schematic frequency trajectories of each stimulus, with varying frequency excursions (amplitude of frequency modulation, $A_{fm}$) and all other parameters fixed at temporal modulation frequency $f_{fm} = 50$ Hz, carrier frequency $f_0 = $ BF (19,187 Hz), $f_{slope} = 0$ Hz/s, duration = 60 ms. Middle, raster responses to stimuli delivered within the vertical green lines. Black dots represent individual spikes. Bottom, mean response tuning curve (black). Frequency excursions smaller than the typical spectral width of pure tone tuning curves drove better responses than the constant pure tone BF itself. This SU had a peak in $A_{fm}$ tuning at 1/10 octave, with an evoked spike rate more than twice that predicted from just integrating the pure tone excitatory tuning curve over the same spectral range. Larger $A_{fm}$ values reduced firing rates from the peak, which would not be explainable just by its excitatory sensitivity to the brief sound's static spectrum. Error bars indicate standard error of the mean (SEM). Spontaneous rate (dotted grey line) and rate predicted from integrating the pure tone tuning curve (grey) are also shown. $*P < 0.01$; $**P < 0.0001$; Bonferroni-corrected $t$ test. Adapted from Chong et al. (2020).

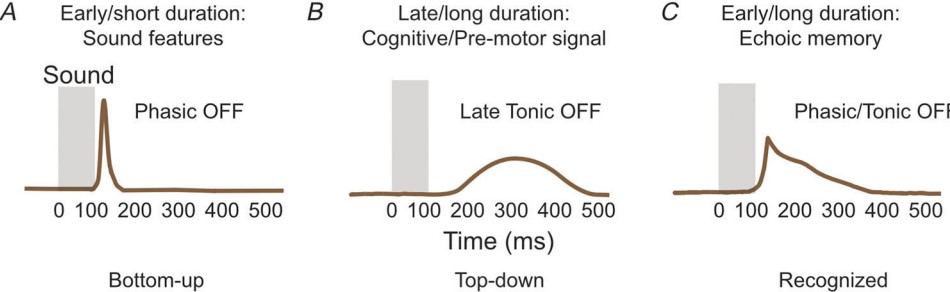

**Figure 4. Three temporal profiles of Off responses in single units**
Neuronal responses in auditory cortex can have at least three different response profiles. *A*, early (short latency) and short duration responses, which most likely are triggered by the sound acoustic features. These phasic Off responses are equivalent to those described in the visual system. *B*, late (long latency) and long duration responses which can be a signature of the sound's significance during learning and/or an efferent copy sent by a motor or premotor area. *C*, early (short latency) and long duration responses can be triggered by an acoustic feature that needs to be maintained in memory during execution of a behavioural task.

