## [Peer Review File · The Journal of physiology]

The What, Which, When, Why and Who of Off Responses in the Auditory System

Jean-Marc Edeline and Robert Liu
DOI: 10.1113/JP289100

Corresponding author(s): Jean-Marc Edeline (jean-marc.edeline@universite-paris-saclay.fr)

Review Timeline:

Submission Date:	16-Apr-2025
Editorial Decision:	12-Jun-2025
Revision Received:	23-Jul-2025
Editorial Decision:	08-Sep-2025
Revision Received:	11-Sep-2025
Accepted:	15-Sep-2025

Senior Editor: Laura Bennet

Reviewing Editor: Laura Bennet

Transaction Report:

Dear Dr Edeline,

Re: JP-TR-2025-289100 "The What, Which, When, Why and Who of Off Responses in the Auditory System" by Jean-Marc Edeline and Robert Liu

Thank you for submitting your manuscript to The Journal of Physiology. It has been assessed by a Reviewing Editor and by 2 expert referees and we are pleased to tell you that it is acceptable for publication following satisfactory revision.

ABSTRACT FIGURES: Authors may use The Journal's premium BioRender account to create/redraw their Abstract Figures (and any other suitable schematic figure). Information on how to access this account is here: <https://physoc.onlinelibrary.wiley.com/journal/14697793/biorender-access>.

REVISION CHECKLIST: Upload a full Response to Referees file. To create your 'Response to Referees' copy all the reports, including any comments from the Senior and Reviewing Editors, into a Microsoft Word, or similar, file and respond to each point, using font or background colour to distinguish comments and responses and upload as the required file type.

We look forward to receiving your revised submission.

Yours sincerely,

Laura Bennet
Senior Editor

REQUIRED ITEMS

- Please include an Abstract Figure file, as well as the Figure Legend text within the main article file. The Abstract Figure is a piece of artwork designed to give readers an immediate understanding of the Review Article and should summarise the main conclusions. If possible, the image should be easily 'readable' from left to right or top to bottom. It should show the physiological relevance of the Review so readers can assess the importance and content of the article. Abstract Figures should not merely recapitulate other figures in the Review. Please try to keep the diagram as simple as possible and without superfluous information that may distract from the main conclusion of the Review. Abstract Figures must be provided by authors no later than the revised manuscript stage and should be uploaded as a separate file during online submission labelled as File Type 'Abstract Figure'. Please ensure that you include the figure legend in the main article file. All Abstract Figures will be sent to a professional illustrator for redrawing and you may be asked to approve the redrawn figure before your paper is accepted.

- Author profile(s) must be uploaded via the submission form. Authors should submit a short biography (no more than 100 words for one author or 150 words in total for two authors) and a portrait photograph of the two leading authors on the paper. These should be uploaded and clearly labelled together in a Word document with the revised version of the manuscript. Any standard image format for the photograph is acceptable, but the resolution should be at least 300 DPI and preferably more. A group photograph of all authors is also acceptable, providing the biography for the whole group does not exceed 150 words.

- It is the authors' responsibility to obtain any necessary permissions to reproduce previously published material and to list these within the main article file. For information, please see: https://jp.msubmit.net/cgi-bin/main.plex?form_type=display_requirements#permissions.

EDITOR COMMENTS

Reviewing Editor:

Both external referees agreed this is a thoughtful and thought-provoking review, which proposes an interesting if still speculative framework for understanding offset responses in the auditory system. Key recommendations emerging from the referee comments are to:

- discuss some additional relevant references;

- mention comparisons to the somatosensory as well as visual modalities; and

- link detailed description of specific unpublished observations (e.g. regarding current injection) more strongly to wider points about the proposed relationship between offset responses and behavioral plasticity.

There are many additional helpful suggestions from the referees for minor revisions.

As editor I suggest one additional minor revision. The penultimate sentence of the conclusion states: "However, if the neural coding studies adopt more unbiased reverse correlation methods (refs), it is clear that Off responses emerge naturally." In fact, this is not the case for neurons with both On and Off responses, because reverse-correlation methods (and most linear-nonlinear models) would not capture excitatory responses to both increases and decreases in intensity well (for the same reason that they do not capture complex cell responses in the visual system). This point is mentioned in the Kopp-Scheinflug et al. 2018 review. Here it is perhaps best to just remove the erroneous statement, as it is not necessary to the conclusion otherwise.

REFeree COMMENTS

Referee #1:

This is an excellent and informative review of the properties and mechanisms of auditory cortical Off-responses. The authors pull together a wide range of aspects and at the end propose a model of the role of two different types of Off-responses. This novel interpretation provides a fertile base for further studies of this often neglected aspect of auditory cortical processing.

I have only a few comments, solely to encourage the authors to include a few additional aspects that may help the readers to better appreciate the relevance of the phenomenon summarized here.

These are a few references whose inclusion into the discussion may help to enhance the review even further.

1) Some more general aspects of Off-set Mechanisms have been discussed in

Anderson LA, Linden JF (2016) Mind the gap: two dissociable mechanisms of temporal processing in the auditory system. *J Neurosci* 36:1977-1995. 10.1523/JNEUROSCI.1652-15.2016

2) Cell-type specific effects of the dynamic behavior of Off-responses is dependent on response history. This publication provides a more detailed investigation of this aspects.

Olsen T, Hasenstaub AR. (2022) Offset Responses in the Auditory Cortex Show Unique History Dependence. *J Neurosci* 42(39):7370-7385.

3) Some implication for functional roles of Off-responses is discussed in these two papers that postulate distinctive aspects of spectral selectivity between Off-response sub populations compared to that of associate On-responses.

Tian B, Kuśmierk P, Rauschecker JP. (2013) Analogues of simple and complex cells in rhesus monkey auditory cortex. *Proc Natl Acad Sci U S A*. 110(19):7892-7.

Sollini J, Chapuis GA, Clopath C, Chadderton P (2018) ON-OFF receptive fields in auditory cortex diverge during development and contribute to directional sweep selectivity. *Nat Commun* 9:1-12. 10.1038/s41467-018-04548-3

Referee #2:

This interesting and thoughtful article reviews the significance of neuronal responses in the auditory system to sound offsets (Off responses). Compared to other sensory modalities (particularly vision), Off responses in the auditory system have, until fairly recently, received little attention. Growing interest in this topic led to the publication of two review articles (Kopp-Scheinpflug et al. 2018 and Anandakumar & Liu, 2022), which respectively focused primarily on subcortical and cortical circuits. In addition to reviewing some of the same material, the present article discusses the evidence for different types of Off response and the importance of behavioral context in interpreting their functional significance. Some parts of the article - particularly the section on current injection parameters and acoustic stimuli evoking Off responses - are likely to attract limited interest outside the field, though this section provides a useful precursor to the later section on behavior and plasticity. Although much of this work is at a relatively early stage (unpublished data are referred to a few times) and many of the conclusions remain speculative, I think the article is suitable for a topical review.

The following are mostly minor comments:

Line 75: "the importance of post-stimulus firing been largely unappreciated in the auditory system". I assume you mean Off responses. While the wording is correct, it might be a good idea to be more explicit here that you are referring to activity following the end of the stimulus as some readers might interpret this as firing following its onset.

Lines 96-99: I'm not sure that the highest standards of objective scholarship are best served by the authors referring to one of their own papers (Anandakumar & Liu, 2022) as "excellent".

Line 173: "...show a dependence to sound duration" should say dependence "on" sound duration.

Line 183: "mechanisms at the origin" => "of" the origin.

Line 200: "repeated ejections of GABA application" should just be "repeated GABA application".

Line 202: I think you mean Fig. 2C-F rather than Fig. 3.

Line 219: "above 50 dB" SPL?

Line 220: "10dB" 10 dB SPL?

Line 222: "were" => was

Line 240: "Inter-Clicks Interval" => Inter-Click interval

Primary auditory cortex is sometimes abbreviated as AI and sometimes as A1 and in other places not abbreviated at all.

Some of the references in the main text include the authors' initials (or first names).

Lines 313-314: This sentence doesn't make sense. Perhaps replace "perform" by "use in".

Lines 325: "it must be the case..." This seems too strong to me - I would recommend saying "it might be the case", particularly as this is a hypothesis.

Line 361: "have" => "has".

Line 405: "these neural activation are"

Line 417: "participate to" => "participate in".

Lines 422-423: "listening natural acoustic streams" Insert "to".

Line 436: "hoped" => "hope".

Figure 1: red is used to depict excitatory inputs in A and inhibitory inputs in B. A more consistent color scheme would be better.

Line 640: "...needs that the" => "requires that the".

Line 641: "On this example, ..." => "In this example,..."

Line 642-643: this slightly confusing sentence would be clearer if the currents were expressed as 4.1-4.3 and 4.6-4.8 nA (rather than using "to").

Line 646: "paring" => "pairing". Delete "ejection".

Line 650: "at the vicinity..." => "in the vicinity".

Figure 2C,D: x-axis title are needed - presumably Frequency (kHz). The axis labels are too small.

Line 657: "center frequency". Perhaps "carrier frequency" would be better to match the term used in Figure 3.

I don't think Figure 3 is cited in the main text (where it is mentioned, the authors mean Fig. 2).

END OF COMMENTS

RESPONSE: We thank the Editor and the Reviewers for their constructive feedback about our review article. We have fully addressed all of the comments and made edits in the manuscript. Details are explained below, point-by-point. Substantive additions to the text now appear in **BLUE**.

EDITOR COMMENTS

Reviewing Editor:

Both external referees agreed this is a thoughtful and thought-provoking review, which proposes an interesting if still speculative framework for understanding offset responses in the auditory system. Key recommendations emerging from the referee comments are to:

- discuss some additional relevant references;

RESPONSE: We thank you for this suggestion. We have now incorporated all the references suggested by the reviewers.

- mention comparisons to the somatosensory as well as visual modalities; and

RESPONSE: We have now included more comparisons to Off responses in the visual system and somatosensory system on bottom of page 4- top of page 5 (in blue), bottom of page 6 (in blue) .

- link detailed description of specific unpublished observations (e.g. regarding current injection) more strongly to wider points about the proposed relationship between offset responses and behavioral plasticity.

RESPONSE: This is an excellent suggestion. We have refined the explanations of the hyperpolarizing current injection study (unpublished) and now explicitly reference that in talking about the behavioral plasticity to USVs, wherein the prevalence of Off-only responses (where firing was inhibited during the sound) is enhanced after learning. These two sentences are in blue in top and bottom of page 13.

There are many additional helpful suggestions from the referees for minor revisions.

As editor I suggest one additional minor revision. The penultimate sentence of the conclusion states: "However, if the neural coding studies adopt more unbiased reverse correlation methods (refs), it is clear that Off responses emerge naturally." In fact, this is not the case for neurons with both On and Off responses, because reverse-correlation methods (and most linear-nonlinear models) would not capture excitatory responses to both increases and decreases in intensity well (for the same reason that they do not capture complex cell responses in the visual system). This point is mentioned in the Kopp-Scheinflug et al. 2018 review. Here it is perhaps best to just remove the erroneous statement, as it is not necessary to the conclusion otherwise.

RESPONSE: Deleted, and the final two sentences now read: "For decades, this methodological bias has led the auditory community to focus just On responses. Instead, we expect that more unbiased views of studying the neural code at play during, but also after, auditory stimuli, will offer deeper insights into how the acoustic and cognitive dimensions of natural sounds are processed in the central auditory system."

REFEREE COMMENTS

Referee #1:

This is an excellent and informative review of the properties and mechanisms of auditory cortical Off-responses. The authors pull together a wide range of aspects and at the end propose a model of the role

of two different types of Off-responses. This novel interpretation provides a fertile base for further studies of this often neglected aspect of auditory cortical processing.

RESPONSE: We thank the Referee for their positive feedback.

I have only a few comments, solely to encourage the authors to include a few additional aspects that may help the readers to better appreciate the relevance of the phenomenon summarized here.

These are a few references whose inclusion into the discussion may help to enhance the review even further.

1) Some more general aspects of Off-set Mechanisms have been discussed in

Anderson LA, Linden JF (2016) Mind the gap: two dissociable mechanisms of temporal processing in the auditory system. *J Neurosci* 36:1977-1995. 10.1523/JNEUROSCI.1652-15.2016

RESPONSE: We now discuss the results of this interesting paper on bottom of page 7.

2) Cell-type specific effects of the dynamic behavior of Off-responses is dependent on response history. This publication provides a more detailed investigation of this aspects.

Olsen T, Hasenstaub AR. (2022) Offset Responses in the Auditory Cortex Show Unique History Dependence. *J Neurosci* 42(39):7370-7385.

RESPONSE: We now discuss this paper in the middle of page 7.

3) Some implication for functional roles of Off-responses is discussed in these two papers that postulate distinctive aspects of spectral selectivity between Off-response sub populations compared to that of associate On-responses.

Tian B, Kuśmierk P, Rauschecker JP. (2013) Analogues of simple and complex cells in rhesus monkey auditory cortex. *Proc Natl Acad Sci U S A*. 110(19):7892-7.

Sollini J, Chappuis GA, Clopath C, Chadderton P (2018) ON-OFF receptive fields in auditory cortex diverge during development and contribute to directional sweep selectivity. *Nat Commun* 9:1-12. 10.1038/s41467-018-04548-3.

RESPONSE: We now mention these two papers on bottom of page 6.

Referee #2:

This interesting and thoughtful article reviews the significance of neuronal responses in the auditory system to sound offsets (Off responses). Compared to other sensory modalities (particularly vision), Off responses in the auditory system have, until fairly recently, received little attention. Growing interest in this topic led to the publication of two review articles (Kopp-Scheinflug et al. 2018 and Anandakumar & Liu, 2022), which respectively focused primarily on subcortical and cortical circuits. In addition to reviewing some of the same material, the present article discusses the evidence for different types of Off response and the importance of behavioral context in interpreting their functional significance. Some parts of the article - particularly the section on current injection parameters and acoustic stimuli evoking Off responses - are likely to attract limited interest outside the field, though this section provides a useful precursor to the later section on behavior and plasticity. Although much of this work is at a relatively early stage (unpublished data are referred to a few times)

and many of the conclusions remain speculative, I think the article is suitable for a topical review.

RESPONSE: We thank the Referee for their positive feedback.

The following are mostly minor comments:

Line 75: "the importance of post-stimulus firing been largely unappreciated in the auditory system". I assume you mean Off responses. While the wording is correct, it might be a good idea to be more explicit here that you are referring to activity following the end of the stimulus as some readers might interpret this as firing following its onset.

RESPONSE: Corrected. Now reads: "Although there has been considerable prior work on explaining how each of these response types have contributed to neural coding in the visual system, until recently the importance of firing after the end of the stimulus has been largely unappreciated in the auditory system."

Lines 96-99: I'm not sure that the highest standards of objective scholarship are best served by the authors referring to one of their own papers (Anandakumar & Liu, 2022) as "excellent".

RESPONSE: Deleted "excellent". (P.S. For the record, this was one of the authors referencing the prior review of the other author.)

Line 173: "...show a dependence to sound duration" should say dependence "on" sound duration.

RESPONSE: Changed.

Line 183: "mechanisms at the origin" => "of" the origin.

RESPONSE: Changed.

Line 200: "repeated ejections of GABA application" should just be "repeated GABA application".

RESPONSE: Changed.

Line 202: I think you mean Fig. 2C-F rather than Fig. 3.

RESPONSE: Corrected.

Line 219: "above 50 dB" SPL?

RESPONSE: Changed.

Line 220: "10dB" 10 dB SPL?

RESPONSE: Changed.

Line 222: "were" => was

RESPONSE: Changed.

Line 240: "Inter-Clicks Interval" => Inter-Click interval

RESPONSE: Changed.

Primary auditory cortex is sometimes abbreviated as AI and sometimes as A1 and in other places not abbreviated at all.

RESPONSE: Corrected.

Some of the references in the main text include the authors' initials (or first names).

RESPONSE: We have corrected this mistake

Lines 313-314: This sentence doesn't make sense. Perhaps replace "perform" by "use in".

RESPONSE: Changed.

Lines 325: "it must be the case..." This seems too strong to me - I would recommend saying "it might be the case", particularly as this is a hypothesis.

RESPONSE: Changed.

Line 361: "have" => "has".

RESPONSE: Changed.

Line 405: "these neural activation are"

RESPONSE: Changed to "activations"

Line 417: "participate to" => "participate in".

RESPONSE: Changed.

Lines 422-423: "listening natural acoustic streams" Insert "to".

RESPONSE: Changed.

Line 436: "hoped" => "hope".

RESPONSE: Based on feedback from the Editor, we changed this last sentence to read, "Instead, we expect that more unbiased views of studying the neural code at play during, but also after, auditory stimuli, will offer deeper insights into how the acoustic and cognitive dimensions of natural sounds are processed in the central auditory system."

Figure 1: red is used to depict excitatory inputs in A and inhibitory inputs in B. A more consistent color scheme would be better.

RESPONSE: While we agree with the Reviewer's point, we note that these figure panels are taken from different journal articles that use different color schemes. Without access to the original figure files, we felt it best to leave the color schemes as is. However, we have added an in-panel legend to identify the different colors of excitation and inhibition in panel B.

Line 640: "...needs that the" => "requires that the".

RESPONSE: Changed.

Line 641: "On this example, ..." => "In this example,..."

RESPONSE: Changed.

Line 642-643: this slightly confusing sentence would be clearer if the currents were expressed as 4.1-4.3 and 4.6-4.8 nA (rather than using "to").

RESPONSE: The sentence has been corrected to read, “When the current was increased to 4.1-4.3 nA, Off responses started to emerge and were even more pronounced when the current was set to 4.6-4.8 nA.”

Line 646: "paring" => "pairing". Delete "ejection".

RESPONSE: Changed.

Line 650: "at the vicinity..." => "in the vicinity".

RESPONSE: Changed.

Figure 2C,D: x-axis title are needed - presumably Frequency (kHz). The axis labels are too small.

RESPONSE: We thank the reviewer for pointing this mistake. The x-axis title was missing and we have fixed this problem.

Line 657: "center frequency". Perhaps "carrier frequency" would be better to match the term used in Figure 3.

RESPONSE: Changed.

I don't think Figure 3 is cited in the main text (where it is mentioned, the authors mean Fig. 2).

RESPONSE: We thank the Referee for catching this. We have now added a reference to Figure 3 in this sentence in Section 2: “Indeed, by fitting the peak frequency of the vocalizations to sinusoidal and linear frequency modulated tones, the authors observed neural tuning around acoustic parameters (especially the amplitude of the frequency modulation, Figure 3).”

END OF COMMENTS

Dear Dr Edeline,

Re: JP-TR-2025-289100R1 "The What, Which, When, Why and Who of Off Responses in the Auditory System" by Jean-Marc Edeline and Robert Liu

Thank you for submitting your manuscript to The Journal of Physiology. It has been assessed by a Reviewing Editor and by 2 expert referees and we are pleased to tell you that it is acceptable for publication following satisfactory minor revision.

ABSTRACT FIGURES: Authors may use The Journal's premium BioRender account to create/redraw their Abstract Figures (and any other suitable schematic figure). Information on how to access this account is here: <https://physoc.onlinelibrary.wiley.com/journal/14697793/biorender-access>.

REVISION CHECKLIST: Upload a full Response to Referees file. To create your 'Response to Referees' copy all the reports, including any comments from the Senior and Reviewing Editors, into a Microsoft Word, or similar, file and respond to each point, using font or background colour to distinguish comments and responses and upload as the required file type.

We look forward to receiving your revised submission.

Yours sincerely,

Laura Bennet
Senior Editor

REQUIRED ITEMS

- Please include an Abstract Figure file, ***as well as the Figure Legend text within the main article file*** (we seem to be missing the legend). The Abstract Figure is a piece of artwork designed to give readers an immediate understanding of the Review Article and should summarise the main conclusions. If possible, the image should be easily 'readable' from left to right or top to bottom. It should show the physiological relevance of the Review so readers can assess the importance and content of the article. Abstract Figures should not merely recapitulate other figures in the Review. Please try to keep the diagram as simple as possible and without superfluous information that may distract from the main conclusion of the Review. Abstract Figures must be provided by authors no later than the revised manuscript stage and should be uploaded as a separate file during online submission labelled as File Type 'Abstract Figure'. Please ensure that you include the figure legend in the main article file. All Abstract Figures will be sent to a professional illustrator for redrawing and you may be asked to approve the redrawn figure before your paper is accepted.

EDITOR COMMENTS

Thank you for addressing the reviewers' and editor's comments so thoroughly. Both reviewers have approved the corrections and changes, with one minor additional correction noted below from Reviewer #1.

Please also see 'Required Items' above.

REFEREE COMMENTS

Referee #1:

The authors did a thorough job in this revision.

I have only one comment:

line 472:'.....to focus just On responses.' should read 'to focus just on On responses.'

To prevent similar confusions it might be better to use capitalized versions referring to ON and OFF activity throughout the text.

Referee #2:

This manuscript has been revised to address all the comments made on the previous version. I have no further comments.

END OF COMMENTS

Dear Senior Editor, Dear Reviewing Editor,

We are pleased to submit a revised version of our review article entitled "*The What, Which, When, Why and Who of Off Responses in the Auditory System*" that should be relevant for the Special Issue on "Mechanism and Function of Stimulus OFF-Responses across Sensory Modalities".

We sincerely thank the Reviewer for detecting a mistake in a sentence on line 472; we have corrected it. We think that it is not necessary to use systematically capitalized versions for referring to ON and OFF activity: we have simply used "On" and "Off" which seems enough to avoid confusions.

We have now a separate file for the graphical abstract legend.

Yours sincerely,

Jean-Marc Edeline and Robert C. Liu

Dear Dr Edeline,

Re: JP-TR-2025-289100R2 "The What, Which, When, Why and Who of Off Responses in the Auditory System" by Jean-Marc Edeline and Robert Liu

We are pleased to tell you that your paper has been accepted for publication in The Journal of Physiology.

Authors should note that it is too late at this point to offer corrections prior to proofing. Major corrections at proof stage, such as changes to figures, will be referred to the Editors for approval before they can be incorporated. Only minor changes, such as to style and consistency, should be made at proof stage. Changes that need to be made after proof stage will usually require a formal correction notice.

Yours sincerely,

Laura Bennet
Senior Editor
The Journal of Physiology

P.S. - You can help your research get the attention it deserves! Check out Wiley's free Promotion Guide for best-practice recommendations for promoting your work at www.wileyauthors.com/eeo/guide. You can learn more about Wiley Editing Services which offers professional video, design, and writing services to create shareable video abstracts, infographics, conference posters, lay summaries, and research news stories for your research at www.wileyauthors.com/eeo/promotion.

IMPORTANT NOTICE ABOUT OPEN ACCESS: To assist authors whose funding agencies mandate public access to published research findings sooner than 12 months after publication, The Journal of Physiology allows authors to pay an Open Access (OA) fee to have their papers made freely available immediately on publication.

You can check if your funder or institution has a Wiley Open Access Account here: <https://authorservices.wiley.com/author-resources/Journal-Authors/licensing-and-open-access/open-access/author-compliance-tool.html>.